



**Forest Types Show Divergent Biophysical Responses After Fire: Challenges to Ecological**

**Modeling**

Surendra Shrestha[1][*], Christopher A. Williams[1], Brendan M. Rogers[2], John Rogan[1], and Dominik

Kulakowski[1]

[1]Graduate School of Geography, Clark University, Worcester, MA 01610

[2]Woods Hole Research Center, Falmouth, MA 02540

*Corresponding Author:

Surendra Shrestha

Graduate School of Geography, Clark University

Worcester, MA 01610

Email: Surshrestha@clarku.edu; sbs.stha111@gmail.com

Phone: +1- (774) 253-0917





**Abstract**
Understanding vegetation recovery after fire is critical for predicting vegetation-mediated
ecological dynamics in future climates. However, information characterizing vegetation recovery
patterns after fire and their determinants are lacking over large geographical extents. This study
uses Moderate Resolution Imaging Spectroradiometer (MODIS) leaf area index (LAI) and albedo
to characterize patterns of post-fire biophysical dynamics across the western United States (US)
and further examines the influence of topo-climatic variables on the recovery of LAI and albedo
at two different time intervals, 10 and 20 years post-fire, using a random forest model. Recovery
patterns were derived for all wildfires that occurred between 1986 and 2017 across seven forest
types and 21 level III ecoregions of the western US. We found differences in characteristic
trajectories of post-fire vegetation recovery across forest types and ecoclimatic settings. LAI in
some forest types recovered only 60% - 70% by 25 years after fire while it recovered 120% to
150% of the pre-fire levels in other forest types, with higher absolute post-fire changes in forest
types and ecoregions that had a higher initial pre-fire LAI. Our random forest results showed very
little influence of fire severity on the recovery of both summer LAI and albedo at both post-fire
time intervals. Post-fire vegetation recovery was most strongly controlled by elevation, with faster
rates of recovery in lower elevations. Similarly, annual precipitation and average summer
temperature had significant impacts on the post-fire recovery of vegetation. Full recovery was
seldom observed when annual precipitation was less than 500 mm and average summer
temperature was above the optimal range i.e., 15-20°C. Climate influences, particularly annual
precipitation, was a major driver of post-fire summer albedo change through its impact on
ecological succession. This study provides quantitative measure of primary controls that could be
used to improve the modelling of ecosystem dynamics post-fire.





Keywords: wildfire; MODIS; post-fire recovery; biogeophysical; remote sensing; succession





## 1. Introduction

Wildfires have burned millions of hectares of forests in the western United States (Littell et al.,
2009; White et al., 2017) and have increased in both frequency and severity in recent decades. This
trend has been attributed to temperature increases, more frequent droughts, below average winter
precipitation and earlier spring snowmelt (Dale et al., 2001; Westerling et al., 2006; Rogers et al.,
2011; Ghimire et al., 2012; Dennison et al., 2014; Littell et al., 2015; Abatzoglou & Williams,
2016; Williams & Abatzoglou, 2016; Williams et al., 2021), making ecosystem resilience and
vegetation recovery post-fire a primary concern to researchers and land managers (Allen &
Breshears, 2015). Existing studies report that large wildfires in western U.S. forests have increased
four-fold since 1970-1986, with total burn area increasing by six and a half times (Westerling et
al., 2006). Expanded burning can profoundly alter a wide range of ecosystem characteristics such
as stand structure, species composition, leaf area, canopy ecophysiology, and microclimate (Liu et
al., 2005). The most immediate biophysical effect of wildfire on the land surface is the decrease in
live vegetation and the deposition of black carbon on the soil surface (De Sales et al., 2018). The
alteration in surface roughness directly influences the interaction between the land and the
atmosphere by, typically, reducing the turbulent mixing and net radiation (Chambers et al., 2005).
Moreover, the deposition of the black carbon on the surface changes net radiation through its
impact on surface albedo, which alters the partitioning of energy into latent heat and sensible heat
(Jin & Roy, 2005). Fires have the potential to modify local to regional climate through these long-
lived changes in land surface dynamics and other substantial forcing impacts such as greenhouse
gas fluxes and aerosols (Bonan et al., 1995). In this study, we use contemporary spaceborne
observing systems to quantify the magnitude and timing of ecosystem responses to severe wildfires
as a crucial step in assessing their associated ecological, hydrological, and biogeophysical impacts.





In addition to quantification, it is equally important to document the factors that determine
variability in post-fire recovery in order to develop a predictive understanding of ecosystem
dynamics in response to wildfire, especially considering present and expected future increases in
the frequency of large, severe wildfires (Scholze et al., 2006; IPCC, 2007; Seastedt et al., 2008;
Urza et al., 2017; Hankin et al., 2019). Vegetation recovery is likely to vary considerably across
the landscape, even when initial estimates of fire severity are similar (Keeley et al., 2008; Frazier
et al., 2018). Some forest ecosystems have shown to recover fully after large severe disturbances
(Rodrigo et al., 2004; Knox & Clarke, 2012), while others have recovered little towards pre-fire
levels (Barton, 2002; Rodrigo et al., 2004; Lippok et al., 2013). Variability in recovery rates has
been shown to depend on the interactive effects of numerous biotic and abiotic factors related to
nature of fire, life history traits of species, and environmental conditions following fire (Chambers
et al., 2016; Johnstone et al., 2016; Steven-Rumann et al., 2018). For example, post-fire recovery
of dry mixed conifer forests in the western U.S. is strongly affected by fire severity (Chappell
1996; Meng et al., 2015; Kemp et al., 2016; Harvey et al., 2016; Meng et al., 2018; Vanderhoof et
al., 2020) and pre-fire condition (Martin-Alcon & Coll, 2016; Zhao et al., 2016). Other factors that
can be important to vegetation recovery after fire include vegetation type (Epting, 2005; Yang et
al., 2017); site topography including slope, aspect, and elevation (Wittenberg et al., 2007; Meng
et al., 2015; Liu et al., 2016; Chambers et al., 2018; Haffey et al., 2018), and post-fire climate
including temperature and moisture conditions (Chappell, 1996; Meng et al., 2015; Stevens-
Rumann et al., 2018; Kemp et al., 2019; Guz et al., 2021). Long-term assessment of post-fire
vegetation recovery across forest types can offer valuable insights to researchers and land
managers who seek to identify areas that could benefit from post-fire management and develop
potential management actions such as fuels treatment, prescribed fire, carbon management, etc.





Several studies have documented vegetation recovery and associated biogeophysical and
biogeochemical dynamics in response to wildfires by employing field-based observations
including flux tower measurements (Chambers & Chapin III, 2002; Jin & Roy, 20005; Amiro et
al., 2006; Randerson et al., 2006; Campbell et al., 2007; Dore et al., 2010; Kemp et al., 2016;
Hankin et al., 2019; Ma et al., 2020), remote sensing observations (Veraverbeke et all., 2012;
O'Halloran et al., 2014; Micheletty et al., 2014; Rogers et al., 2015; Bright et al., 2019; Vanderhoof
et al., 2020), and modeling approaches driven by remote sensing observations (Hicke et al., 2003;
Bond-Lamberty et al., 2009; Williams et al., 2012; Rogers et al., 2013; Maina et al., 2019). While
instructive and critical for mechanistic understanding, local field-based studies on post-fire
ecological dynamics tend to focus on small, localized areas, encompassing only a single or a few
wildfire events (Meigs et al., 2009; Montes-Helu et al., 2009; Downing et al., 2019). In contrast,
large-scale regional analyses using remotely sensed observations and modeling approaches tend
to focus on Mediterranean (Veraverbeke et all., 2012a, 2012b; Meng et al., 2014; Yang et al.,
2017) and boreal ecosystems (Amiro et al., 2000; Chambers & Chapin, 2003; Randerson et al.,
2006; Lyons et al., 2008; Amiro et al., 2010; Jin et al., 2012; Rogers et al., 2013), or on only a few
forest types (mostly ponderosa pine and mixed conifer of western U.S.) (Chen et al., 2011; Dore
et al., 2012; Meng et al., 2015; Roche et al., 2018; Bright et al., 2019). Moreover, such studies
have failed to document how these results scale up to multiple fire events across broad regions.
The purpose of this study is to provide more precise estimate of wildfire impacts on LAI and
surface albedo in seven different forest types of the western US using observations derived from
the MODIS.  Moreover, this study also examines the factors that influence the nature and rate of
vegetation recovery in the post-fire environment. The hypotheses for the work are that 1) the rate
of recovery of LAI following wildfire varies across forest types and ecoclimatic settings, 2) the



change in vegetation cover post-fire induces a change in the albedo which varies by forest types
and ecoclimatic settings, and 3) the variability in the post-fire response of albedo is attributable to
the same factors that explain variability in LAI post-fire.
**2.    Methods**
**2.1. Study Area**
This study was carried out in the western US, a region that has been severely disturbed by wildfires
in the last several decades. Its extent for the purpose of this study (Fig. 1) encompasses the
conterminous US west of the 100th meridian (Thompson et al., 2003). This region is geographically
diverse with high physiographic relief and strong local and regional climatic gradients (Bartlein &
Hostetler, 2003), including regions such as temperate rain forests, high mountain ranges, great
plains, and deserts (Thompson et al., 2003). Our study considered seven forest types that are
dominant across the western US, as defined by the US Forest Service's National Forest Type data
set (Ruefenacht et al., 2008), including Douglas-fir, Pinyon-Juniper, Ponderosa pine,
Spruce/Fir/Hemlock, Mixed conifer, Lodgepole pine, and Oak. Within these forest types, we only
considered areas that were burned with high severity as defined by Monitoring Trends in Burn
Severity (MTBS). Within each ecoregion, we selected only those forest types that cover >10% of
ecoregion's forest area and had >1% pixels burned under high severity. As a result, only 21 out of
35 level III ecoregions of the western US (Table S1) (Omernik, 1987) had a sufficient number of
500 m x 500 m pixels that saw high severity burning within these forest types to support the
generation of forest-type-specific chronosequences of post-fire ecological responses. Across these
ecoregions, average annual precipitation (1981-2010) was $900 \pm 490$ mm yr$^{-1}$ (mean $\pm$ SD), while
mean summer minimum and maximum temperature were $23° \pm 2.8°C$ and $7° \pm 2.5°C$, respectively
(PRISM; Daly et al., 2008).



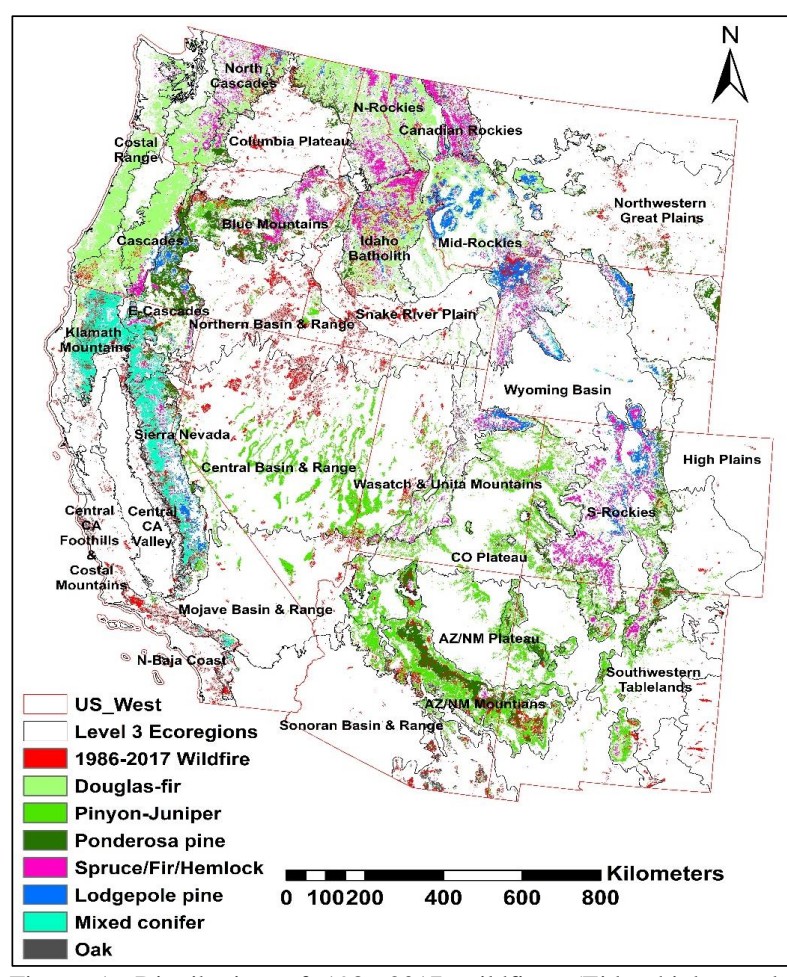

Figure 1: Distribution of 1986-2017 wildfires (Eidenshink et al., 2007) and forest types (Ruefenacht et al., 2008) within study area extent.

## 2.2. Remote Sensing Data and Data Products

The burned area and fire severity data used in this study were obtained from Monitoring Trends in

Burn Severity (MTBS) for the period of 1986-2017 (Eidenshink et al., 2007). We divided our study

into different forest types to analyze the recovery of LAI and albedo post-fire, utilizing a USFS

forest type group map (Ruefenacht et al., 2008). We reprojected the MTBS dataset from its native

30 m resolution to a coarser 500 m resolution. During this process, we retained only those 500 m

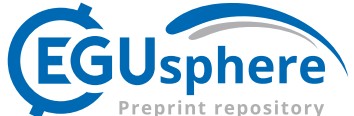

pixels that contained at least 75% of the corresponding 30 m pixels burned, thus reducing noise
from pixels with an unclear mix of burn and unburn conditions. Similarly, we resampled forest
type grid from 250 m to 500 m resolution and selected pixels where at least 75% of the forest
within each pixel belonged to a single forest type based on the 250 m forest type group map. We
excluded pixels that were burned more than once between 1986 and 2017 as such pixels can add
noise to the post-fire trajectory of biophysical properties.
This study analyzed spatially and temporally consistent MODIS products: LAI and shortwave
white sky albedo to assess fire-induced change in vegetation and surface albedo in the western US.
The MODIS satellite data tile subsets (tiles h8v4, h8v5, h9v4, h9v5, h10v4, and h10v5) from 2001
to 2019 were downloaded from the MODIS data archive (https://www.earthdata.nasa.gov/).
Within each data tile, we employed the quality assurance (QA) bits embedded in the MODIS
products to ensure that only the highest-quality values (flagged as '0') were included. This process
involved removing all retrievals affected by cloud cover and those flagged for low quality. The
MODIS LAI product (MCD15A2H; Myneni et al., 2002) reports the green leaf area index which
represents the amount of one-sided green leaf area per unit ground area in broadleaf canopies or
half the total surface area of needles per unit ground area in coniferous canopies. The MODIS LAI
algorithm utilize a main look-up-table (LUT) based procedure that makes use of spectral
information contained in red and NIR bands along with a back-up algorithm that relies on an
empirical relationship between the Normalized Difference Vegetation Index (NDVI) and canopy
LAI, and fraction of photosynthetically active radiation (fPAR) (Myneni et al., 2002).
For albedo, we used the daily MODIS collection 6 bidirectional reflectance distribution function
(BRDF)/Albedo product at 500 m resolution (MCD43A3; Schaaf et al., 2002). The use of both
Terra and Aqua data in this product provides more diverse angular samplings and increased



probability of high input data that allow more accurate BRDF and albedo retrievals. The MODIS
albedo algorithm uses a bidirectional reflectance distribution and shortwave reflectances (0.3-5.0
µm) and provides both black-sky and white-sky albedos. We used shortwave broadband white sky
albedo for this study because it is less biased in complex terrain and less sensitive to view and
solar angles (Gao et al., 2005). We stratified the sampling of white-sky albedo by snow-free and
snow-covered conditions based on the presence or absence of snow, determined at a pixel level by
the MODIS daily snow cover 500 m product (MOD10A1; Salomonson and Appel, 2004). We
assigned snow-free and snow-covered conditions using a threshold of less than 30% and greater
than 75% snow cover. We chose these thresholds as a balance between inclusion for robust
sampling and exclusion to reduce noise from pixels with an unclear mix of snow and snow-free
conditions. We are aware that much of our study domain does not have considerable snow cover
during winter, and these snow-free winter albedos had similar patterns and magnitudes as summer
albedos (Fig. S1). Therefore, the average summer (June-August) albedo values presented here
represent the snow-free condition only, while the average winter (December – February) values
presented include only snow-covered conditions. We did not report winter albedos for all forest
types because of limits on the availability of high-quality snow-covered pixels.
As part of our attribution analysis that seeks to identify factors that influence the pattern of post-
fire biophysical dynamics, we acquired a suite of climate variables– monthly mean summer
precipitation, monthly mean summer temperature, monthly minimum summer temperature,
monthly maximum summer temperature, total annual precipitation– covering the 2001-2019
period from Parameter-Elevation Regressions on Independent Slopes Model (PRISM; Daly et al.,
2008). PRISM utilizes point measurements of precipitation and temperature to generate continuous
digital grid estimations for climate data with a 4 km spatial resolution (Daly et al., 1994). The



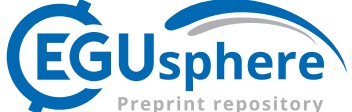

elevation of all burned pixels was taken from the US Geological Survey (USGS) National
Elevation Dataset (NED) at 30 m (U.S. Geological Survey, 2019). All topo-climatic variables were
re-gridded to the 500 m MODIS projection for uniformity.

### 2.3. Generating Chronosequences of Post-fire LAI and Albedo

To address unrealistic variation in MODIS land surface products (Cohen et al., 2006), we
computed mean monthly values by adding all samples and dividing it by the number of samples
in each month within our stratified design. For the summer season, we computed mean yearly
values of LAI and albedo by averaging the data from June, July, and August. Similarly, for the
winter season, yearly values of LAI and albedo were computed the same way using data from
December, January, and February. Next, we analyzed changes in post-fire LAI and albedo relative
to pre-fire by sampling each of them as an annual time series from three years before wildfire
events to all years of record after wildfire events. We grouped samples from each fire event based
on forest type, eco-climatic setting, and snow cover conditions. Within these groups, we
composited burn events from different years and aligned them temporally to represent three years
prior to the fire and all years after the fire. Consequently, chronosequences of biophysical
properties as a function of time since fire were created for a combination of seven forest types, two
snow cover conditions (in case of albedo), and 21 sub-ecoregions.

### 2.4. Attribution of Recovery

We explored the relationships between albedo and LAI recovery and topo-climatic factors, and
subsequently attributed the recovery at 10 years post-fire and 20 years post-fire using random
forest (RF) algorithms, implemented in R (Breiman 2001; Liaw & Wiener, 2002). We used a non-
parametric modeling method because most variable distributions were non-normal and RF does



not require the variables to be normally distributed. Additionally, RF can handle tens of thousands
of data points and provides variable importance scores. We initially selected seven explanatory
variables - fire severity class (low, medium, and high), three temperature variables, two
precipitation variables, and elevation. Although RFs do not require collinear variables to be
removed (Breiman, 2001), we employed a Variance Inflation Factor (VIF) analysis for
multicollinearity as a variable selection method to improve computation efficiency and enhance
interpretation, particularly with respect to variable importance. VIF analysis involves: a)
calculating VIF factors, b) removing the predictors from this set with VIF>10, and c) repeating
until no variable has VIF>10. This provided us with four uncorrelated predictors to be used in the
RF model - fire severity class, total annual precipitation, mean summer temperature (June –
August), and elevation. We pooled post-fire LAI and albedo responses across 21 ecoregions within
a given forest type for both time-intervals (10-year post-fire and 20-year post-fire). The dataset
was divided into training (80%) dataset to train the RF model and test (20%) dataset to validate
the model. We created four RF models for each forest type (one for each time interval for both
LAI and albedo) using fire and topo-climatic variables to determine how fire severity, climate and
topography variables contributed to the recovery of summer LAI and albedo at two different times
after burning- 10 years post-fire and 20 years post-fire. We tuned the model to generate a model
with the highest accuracy i.e., the lowest out-of-bag error among all tested combination of
parameter values. The model's performance was assessed using the $R^2$ metric. We used unscaled
permutation accuracy instead of the traditional Gini-based importance metric to rank the relative
importance among explanatory variables, as Gini-based importance was shown to be more strongly
biased towards continuous variables or variables with more categories compared to other
importance metrics (Strobl et al., 2007). The unscaled permutation importance metric calculates





variable importance scores as the amount of decrease in the accuracy when a target variable is
excluded. We used partial dependence plots (PDP) to visualize the influence of each explanatory
variable on the degree of 10 years and 20 years post-fire recovery of LAI and albedo. PDP
quantifies the marginal effects of a given variable on an outcome and provides a mechanism to
explore insight in big datasets, especially when the random forest is dominated by lower-order
interactions (Martin, 2014).

## 224    3. Results

### 225    3.1. Post-fire Recovery of Land Surface Properties

Burning caused a large decline in LAI for all forest types. Generally, high productivity forests
(e.g., Douglas-fir and Mixed conifers), compared to other forest types, experienced a larger decline
in LAI in year one after fire (Fig. 2a-g). Compared to pre-fire levels, the decline in LAI ranged
from 47% in Pinyon-Juniper to 76% in Ponderosa pine forests (Table S2). After this initial
decrease, the effects of vegetation regeneration became apparent. For all forest types, the
magnitude of LAI change decreases with increase in time since fire. However, LAI did not recover
to the pre-fire condition in most cases by the 25-year period of observation available for this study.
We found large differences in the timing of LAI recovery across forest types, with forest types
recovering at different rates, crossing the pre-fire levels at different times, and reaching different
peaks in LAI (Fig. 2a-g). For example, Douglas-fir in Columbia Mountains, Klamath Mountains,
and Southern Rockies (Fig. 2g) and Mixed conifers in Baja California and Eastern Cascades (Fig.
2a) showed complete recovery of LAI to pre-fire levels within the 25-year study period, while
Lodgepole pine, Oak, and Ponderosa pine were characterized by a slower recovery rate and most
did not recover to pre-fire levels by the 25-year period (Fig. 2 and Table S2). We also found varied
recovery rates across geographic regions even within a single forest type, presumably related to





climate and soils. For example, the characteristic post-fire LAI trajectories for the high
productivity Douglas-fir forest type (Fig. 2g) showed a substantially faster recovery in Cascades,
Klamath Mountains, and Columbia Mountains regions compared to the Idaho Batholith region of
the western US. Based on observations from all forest types, in general, the faster recovery of LAI
was observed in high wet areas with substantial maritime influences.

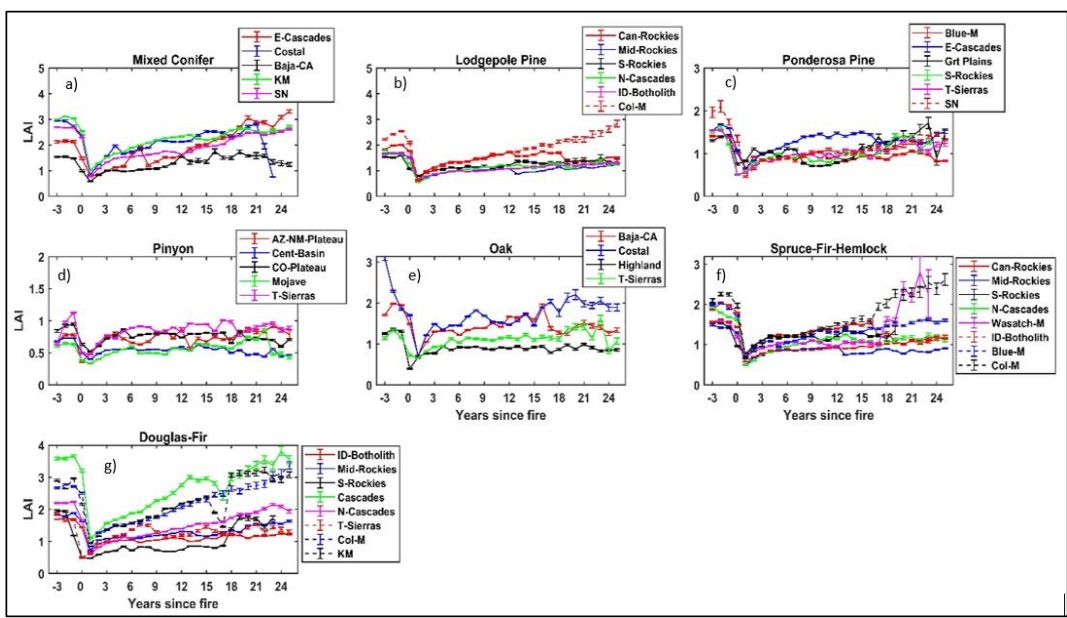


Figure 2: Mean summer post-fire LAI (± SE) as a function of time since fire in seven different
forest types of the western US. (Sub-ecoregions: E-Cascades = Eastern Cascades; Costal = Costal
sage; Baja-CA = Baja California; KM = Klamath Mountains; SN = Sierra Nevada; Can-Rockies
= Canadian Rockies; Mid-Rockies = Middle Rockies; S-Rockies = Southern Rockies; N-Cascades
= Northern Cascades; ID-Batholith: = Idaho Batholith; Col-M = Columbia Mountains; Blue-M =
Blue Mountains; Grt Plains = Great Plains; T-Sierras = Temperate Sierras; AZ-NM-Plateau =
Arizona-New Mexico Plateau; Cent-Basin = Central Basin; CO-Plateau = Colorado Plateau;
Mojave = Mojave Basin; Highland = North American Highland; Wasatch-M = Wasatch
Mountains).
Turning to albedo, we found significant changes in summer albedo post-fire of all forest types.
Three important trends, similar among forest types, emerged from these post-fire summer albedo
trajectories. First, for all forest types, summer albedo decreased immediately after fire (Fig. 3)





likely due to low reflectivity by black carbon deposition on the soil surface and dead tree boles
both common immediately after high severity burning. The decline in summer albedo ranged from
0.01-0.02 across forest types with the greatest decline (20% from pre-fire levels; Table S3)
observed in Douglas-fir forest of the Klamath Mountains region. Second, post-fire albedo
increased gradually from year two since fire, crossing the pre-fire levels at around 3 years post-
fire, and peaking at different time intervals for different forest types and regions (Fig. 3a-g).
Elevated post-burn albedo is presumably due to increasing canopy cover, the relative high albedo
of grasses and shrubs that establish in early succession, and the loss of black carbon coatings on
soil and woody debris (Chambers and Chapin, 2002). The timing and magnitude of peak post-fire
albedo varied across forest types. For example, Ponderosa pine showed its peak in post-fire albedo
at 18 years post-fire (Fig. 3c) and 11 years post-fire for one of the Mixed Conifer regions (Fig. 3a),
while slow growing species such as Spruce/Fir/Hemlock may not have reached its peak by the end
of the 25-year post-fire study period (Fig. 3f). Similarly, we see significant regional differences in
timing and magnitude of peak for a given forest type group. For example, Mixed Conifer post-fire
albedo peaked at 11 years post-fire in Baja California, while it continued to increase through to 25
years in Klamath Mountains (Fig. 3a). Third, as the post-fire LAI approached the pre-fire LAI
levels, post-fire albedo started to decline from the peak towards its pre-fire albedo, but it did not
reach the pre-fire albedo levels by the end of the 25-year study period (Fig. 3a-g). Post-fire winter
albedo for each forest type had a similar pattern as summer albedo except with greater magnitude
and that it increased immediately after fire (Fig. 4a-f and Table S4). We observed greater inter-
annual variability in the timeseries of post-fire winter albedo likely related to greater noise
associated with variability in snow cover and also smaller sample sizes. The albedo response was
more than three-fold larger in winter than in summer, peaking in the range of 0.4 to 0.6 across





forest types and with an increase over pre-fire levels of about 0.25 to 0.50. Similar to summer
albedos, winter albedos did not return to the pre-fire levels by the end of 25-year study period (Fig.
4a-f).

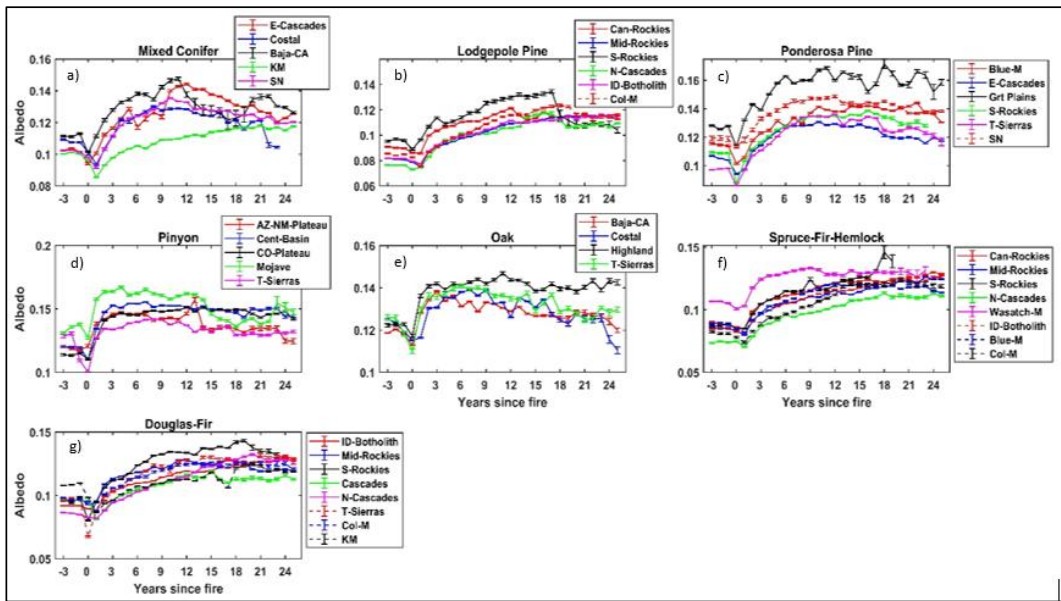

Figure 3: Mean summer post-fire albedo (± SE) as a function of time since fire in seven different
forest types of the western US.





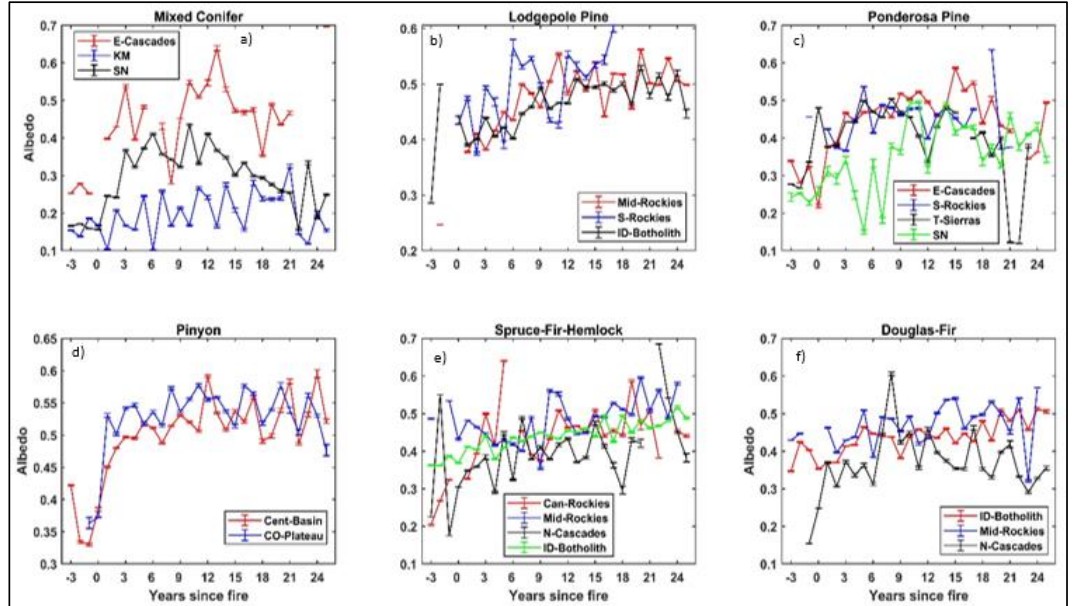

Figure: 4: Mean winter post-fire albedo (± SE) as a function of time since fire in seven different forest types of the western US.

## 3.2. Drivers of post-fire recovery of LAI and albedo

Our random forest model had high accuracy for recovery of both LAI and albedo 10 years and 20 years post-fire. The out-of-bag (OOB) error rate of the random forest model for the relative recovery of 10-year post-fire LAI was around 3% - 8% ($r^2 = 0.66 – 0.78$), while it was around 2.5% - 9% ($r^2 = 0.65 – 0.78$), 0.4% - 1.4% ($r^2 = 0.55 – 0.83$), and 0.3% - 1.6% ($r^2 = 0.52 – 0.83$) for 20-year post-fire LAI, 10-year post-fire albedo, and 20-year post-fire albedo, respectively (Table S5). The variable with greatest importance agreed well between 10-year LAI and 20-year post-fire LAI for all forest types indicating that the recovery of LAI at 10-year and 20-year post-fire were both largely determined by the same governing factors (Fig. S2). Among all the explanatory variables, the degree of post-fire LAI recovery at both 10-year and 20-year post-fire were largely dominated by elevation and total annual precipitation (Fig. S2). In contrast, the factor with greatest influence on post-fire summer albedo varied by forest type and time since fire. For



example, in the Mixed conifer forest type, annual precipitation was the major determinant of 10-
year post-fire albedo recovery, while it was average summer temperature in case of 20-year
postfire. Similarly, the degree of 10-year post-fire albedo recovery in the Spruce/Fir/Hemlock
forest type was largely determined by average summer temperature, while the recovery after 20-
year post-fire was mainly determined by elevation. Fire severity, on the other hand, showed almost
no explanatory power in predicting recovery of LAI and albedo at both times for all forest types
(Fig. S2,S3).
The degree of LAI recovery 10-year post-fire increased with an increase in total annual
precipitation for all forest types, but it varied little when the total annual precipitation exceeded
1000 mm. Annual precipitation was the major determinant of 10-year postfire LAI recovery for
dry forests like Ponderosa pine, Pinyon-Junipers, and Oak, and these forest types tended to recover
above pre-fire levels as the annual precipitation is increased. However, when the annual
precipitation is less than 500 mm, the relative change in LAI is below 0 for all forest types,
indicating that the complete recovery of LAI 10-year postfire was unlikely with annual
precipitation less than 500 mm (Fig. 5c). In contrast, five out of seven forest types recovered over
pre-fire levels 20-years post-fire with increased annual precipitation, indicating that Mixed
conifers and Douglas-fir need more time and higher annual precipitation to recover to the pre-fire
level. Only Oak and Ponderosa pine showed increased LAI 20-year post-fire as the annual
precipitation exceeded 2000 mm (Fig. 6c). As with LAI, annual precipitation was one of the major
determinants of both 10-year and 20-year post-fire albedo recovery. The post-fire elevation of
albedo by 10 years was larger for sites with less annual precipitation (Fig. 7c and 8c), particularly
noticeable in dry forest types such as Douglas-fir, Ponderosa pine, and Oak where increased
precipitation triggered a rapid increase in post-fire vegetation recovery. The Oak forest type




showed a particular anomaly of albedo 20-years post-fire, exhibiting a decline of around 20%
below pre-fire levels for sites with annual precipitation of 2000 mm or above (Fig. 8c), consistent
with a rapid increase in vegetation recovery.
Regarding average summer temperature, we found interesting divergence in the pattern of LAI
response between cool and hot climates. For forests growing in hotter conditions, the magnitude
of LAI recovery at both time intervals decreased in areas with higher temperatures, particularly in
Oak, Pinyon-Junipers, and Ponderosa pine forest types, as these forest types grow at warmer end
of the species distribution. In contrast, increases in average summer temperature assisted the
recovery of forest types growing at the colder end of the species distribution such as Lodgepole
pine and Spruce/Fir/Hemlock (Fig. 5d and 6d), noting that LAI was consistently lower than pre-
fire levels for these forest types at both time intervals. Albedo does not show the same divergence
in pattern with warmer conditions, and instead we find a somewhat surprising pattern. Hotter sites
tend to see a larger elevation of summertime albedo over the pre-fire condition at both time
intervals in spite of faster recovery of LAI with hotter temperature (Fig. 7d and 8d).
Elevation was consistently found to be an important variable in determining the trajectory of post-
fire vegetation recovery. The post-fire recovery of LAI was slower at higher elevation both 10-
years and 20-years post-fire. Most forest types showed complete recovery towards pre-fire levels
at an elevation below 1500 m. Only Pinyon-Junipers and Ponderosa pine forest types saw faster,
more complete recovery of LAI with higher elevation (Fig. 5b and 6b). Turning to albedo response,
we found that higher elevation led to a smaller increase in albedo over its pre-fire value for both
time periods for the two forest types for which elevation was the most important predictor of post-
fire albedo change, namely for Pinyon-Juniper and Ponderosa pine forests. This is consistent with
faster post-fire recovery of LAI at higher elevation portions of range for these two forest types. In





contrast, post-fire albedo of Douglas-fir, Mixed conifer and Oak forest types showed little
dependence on elevation (Fig. 7b and 8b).
Although fire severity was the least important predictor of both post-fire LAI and albedo recovery
at both time events, our results showed significant variation in post-fire recovery among severity
classes for all forest types. As expected, the overall recovery of LAI 10-year post-fire was greater
for low fire severity where the recovery ranged between 85% and 95% of pre-fire LAI levels (Fig.
5a). Only in the case of Oak and Pinyon-Juniper forest types that burned with high severity did we
see full recovery of LAI at or above pre-fire levels by 10-years post-fire. By 20 years post-fire,
Lodgepole pine and Spruce/Fir/Hemlock still show a suppression of LAI relative to pre-burn and
less recovery for more severe burn conditions (Fig. 6a) while Oak sees LAI elevated over the pre-
burn condition and saw the largest LAI at sites that had the highest severity fires (Fig. 6a).  The
four other forest types had LAI equal to the pre-burn condition and showed no variation across fire
severity. For albedo, all forest types showed a larger elevation of albedo over their pre-fire values
under medium fire severity (Fig. 7a). Oak had the lowest change in albedo at both time events,
owing to rapid post-fire recovery. Overall, post-fire albedo was consistently higher than pre-fire
levels at both time events in all forest types indicating that albedo requires more than two decades
to return to pre-fire levels in these forest types (Fig. 7a and 8a).

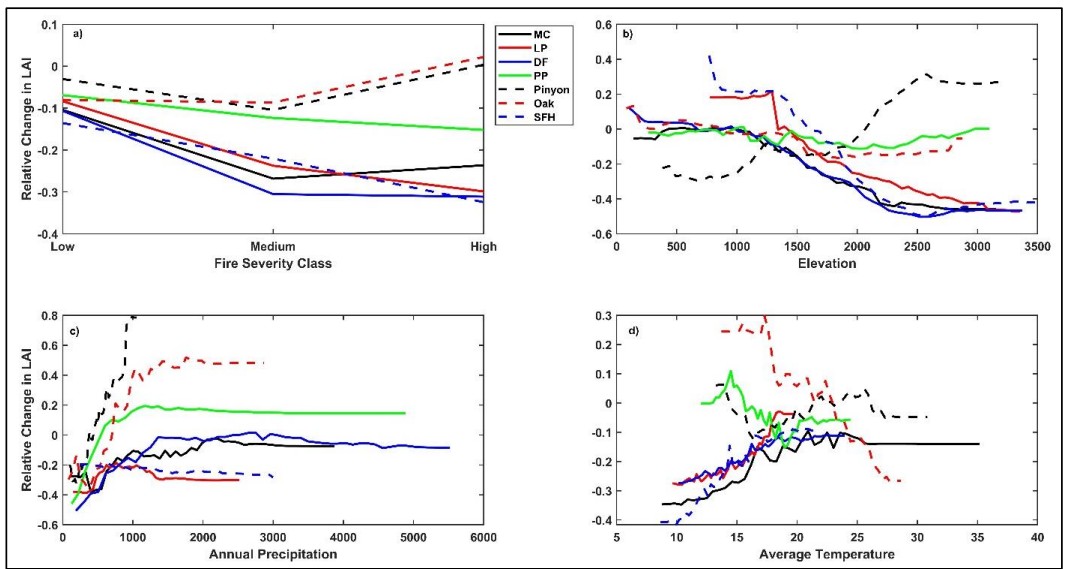

Figure 5: Partial dependence of change in summer LAI 10-year post-fire relative to pre-fire on a)
fire severity, b) elevation, c) annual precipitation, and d) mean monthly summer temperature.
(Forest types: MC = Mixed Conifers; LP = Lodgepole pine; DF = Douglas-fir; PP = Ponderosa
pine; Pinyon = Pinyon-Juniper; SFH = Spruce/Fir/Hemlock). The y-axis represents change in LAI
post-fire relative to pre-fire (degree of recovery), where negative values represent recovery below
pre-fire levels, 0 represents recovery to pre-fire levels, and positive values represent recovery
above pre-fire levels.

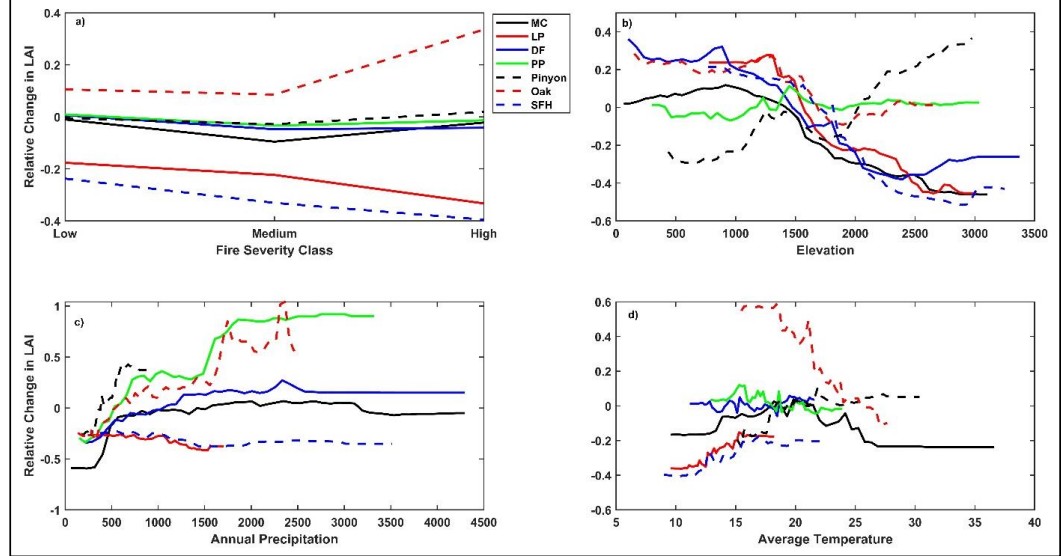

Figure 6: Partial dependence of change in summer LAI 20-year post-fire relative to pre-fire on a)
fire severity, b) elevation, c) annual precipitation, and d) mean monthly summer temperature.

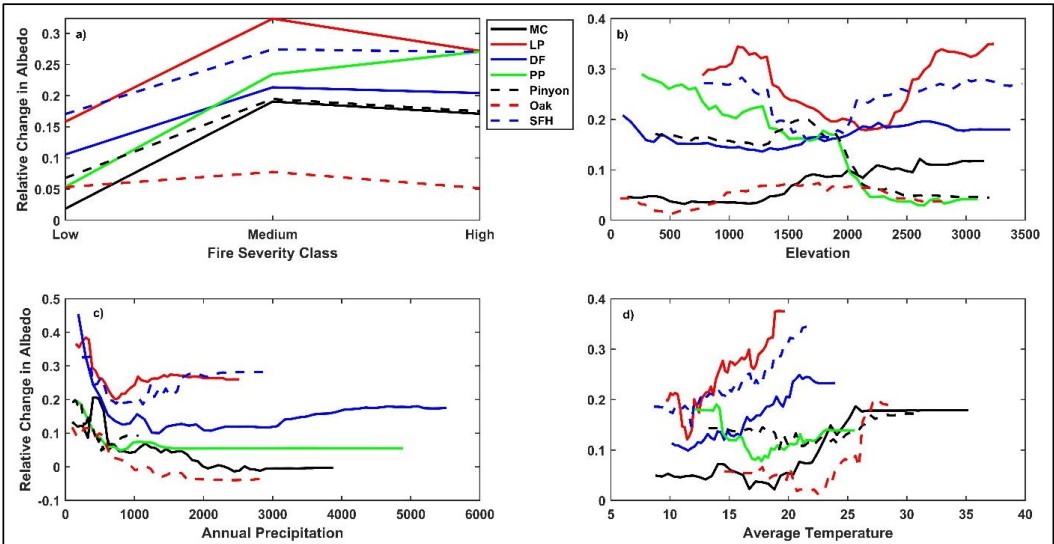

Figure 7: Partial dependence of change in summer snow-free albedo 10-year post-fire relative to pre-fire on a) fire severity, b) elevation, c) annual precipitation, and d) mean monthly summer temperature. The y-axis represents change in albedo post-fire relative to pre-fire (degree of recovery), where negative values represent recovery below pre-fire levels, 0 represents recovery to pre-fire levels, and positive values represent recovery above pre-fire levels.

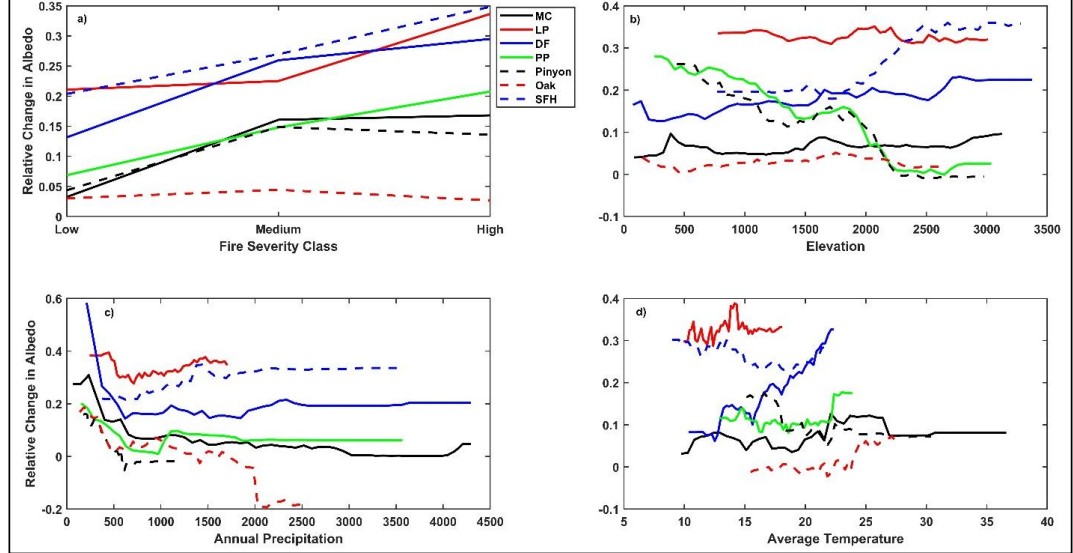

Figure 8: Partial dependence of change in summer snow-free albedo 20-year post-fire relative to pre-fire on a) fire severity, b) elevation, c) annual precipitation, and d) mean monthly summer temperature.

## 4. Discussion and Conclusion



Here, we extended the regional research by Shrestha et al., (2022) with a much broader sampling
to study post-fire responses for seven forest types in 21 sub-ecoregions of the western U.S. In
addition, this study also uses a machine learning approach (random forest) to examine the influence
of several topo-climatic variables on the nature and rate of vegetation recovery and associated
albedo in the post-fire environment.
**4.1. Post-fire Vegetation Recovery**
In this study, we used MODIS-derived LAI to increase our understanding of variability in the
recovery of vegetation in the post-fire environment across seven forest types and 21 sub-
ecoregions of the western United States. Our study focused on the change of LAI over 25 years
post-fire. During this timeframe, the recovery of LAI to the pre-fire condition can be expected to
reflect establishment of new vegetation as well as the (re)growth and expansion of vegetation that
managed to survive the wildfire. Similar to other studies (Morresi et al., 2019; Vanderhoof et al.,
2020), we found rapid vegetation recovery in the first 10 years after fire. While LAI rebounded
rapidly in the initial 10 years post-fire, this cannot be taken as a definitive indicator of successional
trajectory, especially for slow growing forests like subalpine fir (Ferguson and Carlson, 2010) or
for forests with episodic post-fire germination such as Ponderosa pine (Savage et al., 1996; Brown
and Wu, 2005; Rodman et al., 2019). Leaf area recovery then slowed in most cases, and for many
it did not return to the pre-fire level by the end of study period. We anticipate that the recovery of
LAI to its pre-fire condition continues to unfold over time, extending beyond the 25-year duration
covered by our study. In some cases, we see LAI at 20 or 25 years post-fire exceeding that prior
to burning, suggesting that wildfire may have stimulated canopy renewal or release of the
understory. Evaluating post-fire LAI trajectories on these, and longer, timescales can be of value



from a management perspective, for example, to identify regions where there is a risk of
regeneration failure for dominant, native species (Welch et al., 2016).
Our findings generally agree with basic biogeographic expectations. For example, differences in
characteristic trajectories exist across forest types and ecoregions related to climate as well as soils
and the basic fire adaptation traits of the species. Fire caused a similar proportional reduction of
LAI across forest types and ecoregions, generally with 30% to 70% reduction in year 1 post-fire
but with smaller reductions in some Pinyon-Juniper setting. Correspondingly, the absolute
magnitude of LAI decline caused by fire was larger in forest types and regions that had a higher
initial pre-fire LAI (Table S2). We also found varied rate of LAI recovery post-fire across forest
types and ecoregions. Some forest types saw recovery to only 60 % to 70% by 25 years while
others saw LAI recovery to 120% to 150% of the pre-fire condition (Table S2). Similar to decline
in LAI year 1 post-fire, the absolute value of LAI increases 25 years post-fire was larger in settings
that had a larger pre-fire LAI, meaning in eco-climatic settings that are relatively favorable for
forest growth. Many factors are likely to contribute to these patterns across forest types and
ecoclimatic settings. First and foremost, it is no surprise that areas more suitable for growth have
faster and more complete recovery with higher absolute LAI within a given forest type. For
example, Douglas-fir stands in Cascades, Columbia Mountains, and Klamath Mountains had faster
recovery rates and greater changes in absolute LAI after year 1 post-fire than did stands in the
Rockies and Temperate Sierras (Table S2). Similarly, we observed a consistent slow trend in the
rate of conifer regeneration in the interior of the western US with continental climate where high
severity fire is common. This is likely because much of dry montane conifers and subalpine forests
in the east of North Cascades, compared to western side, are characterized by higher proportion of
high severity burn patches during dry years, and as the fires get larger, the interior area of the burn



patches increases significantly resulting in reduced establishment rates due to reduced seed
availability (Cansler and McKenzie, 2014). While we did not examine the evidence of seed
availability being a limitation for LAI recovery post-fire, it may become a growing limitation in
these forests with wildfire becoming more severe in recent decades (Westerling et al., 2006; Parks
and Abatzoglou, 2020) and the likely increase in persistent burned patch density under more
extreme fire weather condition (Krawchuk et al., 2016). The regeneration capacity of the dominant
tree species post-fire is also likely to play a role, with some readily and actively resprouting or
having serotiny, while other lack these fire-adaptation traits (Howard, 2003; Meng et al., 2018)
that can be important for ecological resilience. Post-fire regeneration may also be impacted by
secondary factors like competition with other species such as early colonizers common after
burning. This is particularly true in Ponderosa pine and Lodgepole pine stands as these species can
be outcompeted by aspen over the first 10-15 years postfire (Hansen et al., 2016; Stoddard et al.,
2018; Vanderhoof et al., 2020).  The post-fire dynamics presented here are not stratified by post-
fire species composition, only characterizing the biophysical characteristics that unfold after
burning of a particular forest type. Naturally, post-fire species composition can differ from pre-
fire depending on seed and nutrient availability, fire severity, and climate and these effects are
embedded in the post-fire biophysical trajectories that we present. Further exploration of how post-
fire species composition and other regeneration characteristics influence biophysical trajectories
is warranted.
Our findings of post-fire LAI trajectories across ecoclimatic settings suggest that Douglas-fir
stands may be less vulnerable to climate warming compared to Ponderosa pine, as their current
range tends to extend into cooler and moisture areas where they recover above pre-fire levels
within 25 years post-fire. This indicates that the worsening of climate changes in the future (more





periods of prolonged drought) can have implications for migration of ponderosa pine due to
worsening regeneration under climate stress. Such fire-catalyzed vegetation shift in coming years
to decades can significantly affect the ecosystem services and economic activities provided by
these widespread forest types (Rogers et al., 2011; Coop et al., 2020); thus, it is critically important
to gain a comprehensive understanding of how the ranges of species may expand as tree growth
becomes more feasible in higher elevations and higher latitudes (Lenoir et al., 2008) for forest
management of burned areas in coming decades. Although Pinyon-Juniper forests recovered
rapidly in the first few post-fire years, our observed decline in the rate of pinyon-juniper recovery
is consistent with the findings of Vanderhoof et al., (2020). This forest type is recognized for its
slow regeneration and susceptibility to drought (Hartsell et al., 2020). Existing studies in post-fire
recovery of Pinyon-Juniper suggest that this forest type recovers to pre-fire condition in <5 years
after fire in the case of low to moderate fire (Jameson, 1962; Dweyer and Preper, 1967), while it
takes >100 years for recovery to pre-fire condition under high severity with heavy Pinyon-Juniper
mortality (Erdman, 1970; Koniak, 1985). Other forest types showed faster or similar rates of
recovery, for instance, Mixed conifer recovered completely in most of the ecoregions of the
western US possibly due to richer species diversity and relatively higher precipitation (Bright et
al., 2019).
**4.2. Post-fire albedo Changes**
Our results provide evidence for significant effects of wildfires on the albedo across forest types
and eco-climatic settings in the western US, with post-fire albedo being much higher albedo in
winter than in summer. The post-fire albedo trajectories obtained from this study are broadly
consistent with those obtained from the literature (Beringer et al., 2003; Randerson et al., 2006;
Lyons et al., 2008; Montes-Helu et al., 2009; Gleason et al., 2019). All forest types showed



noticeable age-dependent albedo patterns, with a transient peak in summer albedo around 10-18
years post-fire. We observed a decline in summer albedo during the first year after fire except for
Pinyon-Juniper (Table S3) from charred surface and the deposition of black carbon. The increase
in albedo in first year after fire in Pinyon-Juniper may be associated with low pre-fire LAI leading
to lower levels of charcoal and black carbon deposition that absorb incoming radiation. Our finding
is comparable to previously published findings that report albedo drops in the range of 0.01-0.05
using MODIS albedo (Jin and Roy, 2005; Randerson et al., 2006; Lyons et al., 2008; Veraverbeke
et al., 2012). The slight differences are likely related to the variability in the domain of each study
(e.g., western US vs. boreal, western US vs. Mediterranean), spatial resolution of MODIS pixels
(500 m) that includes unburned patches and non-forest fractions, illumination conditions of the
MODIS albedo products (black sky, white sky, blue sky) and method used to calculate albedo
differences. Regarding the latter, we compared a pixel to itself between pre-and-post-fire years.
The approach of comparing burned pixels to unburned neighboring pixels as control is also
common (e.g., Myhre et al., 2005; Randerson et al., 2006; Lyons et al., 2008; Gatebe et al., 2014).
One issue with this approach is that it does not consider heterogeneity of the land surface. Burned
and control pixels may not be equivalent in the pre-burn period (Dintwe et al., 2017), as they do
not necessarily represent a comparable vegetation state and therefore may not be a good proxy to
pre-fire state. This characteristic decline in summer albedo immediately after fire contributed to
differences in albedo patterns with other disturbance types (harvest, beetle outbreak). For example,
in the first year following a disturbance event, Mohammad et al., (2019) reported higher summer
albedo in a post-harvest stand than in a post-fire stand because of high charcoal occurrence on the
soil surface in the latter case.



Soon after fire, we observed an increased in post-fire albedo during the summer period due to
combination of char removal and presence of early-successional plants (Johnstone et al., 2010)
that have higher albedo than mature species (Betts and Ball, 1997; Pinty et al., 2000; Amiro et al.,
2006; Dintwe et al., 2017). Summer post-fire albedo recovered faster than LAI regardless of
vegetation type. This pattern suggests that, in contrast to findings of Pinty et al., (2000) and
Tsuyuzaki et al., (2009), post-fire recovery of albedo is driven by multiple factors in addition to
the early regeneration of vegetation such as vegetation destruction and charcoal left behind (Jin et
al., 2012), differences in fuel combustion and consumption (Jin and Roy, 2005), species
composition during early succession (Beck et al., 2011), and seasonal variation in soil moisture
and removal of black carbon (Montes-Helu et al., 2009; Veraverbeke et al., 2012). As the
regenerating vegetation matures, the increase in post-fire albedo progressively weakens as
suggested by Amiro et al., (2006), reaching peak at ~ 10-18 years post-fire which then gradually
decline towards pre-fire levels. We did not observe the complete recovery of post-fire albedo
within the study period of 25 years post-fire. Many studies using remote sensing technique suggest
that albedo in post-fire stands commonly equilibrates at ~40-80 years post-fire (Randerson et al.,
2006; Lyons et al., 2008; Kuusinen et al., 2014; Bright et al., 2015; Mohammad et al., 2019, Potter
et al., 2020).
We found the greatest increase in post-fire albedo during winter, a finding consistent with others
(Liu et al., 2005; Randerson et al., 2006; Montes-Helu et al., 2009; Gleason et al., 2019) due to
increased exposure of snow resulting from the loss of canopy and tree mortality. In our analysis,
post-fire winter snow-covered albedo increased with time since fire until a peak was reached, the
timing of which varied across forest types. We hypothesize that this increase with time may result
from the fall of standing dead snags (O'Halloran et al., 2012) and lower rate of reestablishment



during succession (Fig. S4). On average, it takes 5-15 years after fire for half of the dead snags to
fall in post-fire environment in coniferous forests in western North America (Russell et al., 2006),
which coincides with the timing of peak in winter albedo in our study. Our finding showed similar
post-fire winter albedo pattern across forest types in a region. For example, winter albedo in
Lodgepole pine, Spruce/Fir/Hemlock, and Douglas-fir forest types in the Idaho Batholith region
increased at a similar rate with time since fire which corresponds to consistent lower LAI recovery
rate across these forest types in this region (Fig. S4b,f,g) related to climate and soil. However,
variation in winter albedo was greater across ecoregions within a forest type (e.g., Mixed conifer)
owing to variable rates of post-fire LAI recovery (Fig. S4a). Overall, our findings indicate a strong
dependency of post-fire seasonal albedo on the proportion of vegetative cover, irrespective of
forest types, on the post-fire environment. This observed effect provides a strong connection
between albedo and successional patterns observed in these specific forest types.
**4.3. Controls on post-fire recovery of biophysical parameters**
One of the major contributions of our approach is that it not only generates the post-fire trajectories
of land surface biophysical properties across a range of forest types and geographic regions, but
also distinguishes the contribution of nature of fire, climate, and topography on post-fire LAI and
albedo recovery for each forest type. Previous work has shown fire severity to be an important
driver of regeneration, with high fire severity associated with lower post-fire regeneration
(Crotteau et al., 2013; Meng et al., 2015; Chambers et al., 2016; Vanderhoof et al., 2020). In
contrast, our analysis suggested fire severity was of relatively low importance relative to other
variables considered (Fig. S2). We found that higher rates of post-fire recovery were associated
with low severity fire and lowest recovery rates were associated with high fire severity. The lower
recovery rates associated with high fire severity are possibly due to lower seed availability and



greater distance to live seed sources (Haire & McGarigal, 2010; Kemp et al., 2016; Kemp et al.,
2019), but high fire severity can also create mineral seed beds and free up essential resources such
as moisture, light, and nutrients which promote the growth of vegetation (Gray et al., 2005;
Moghaddas et al., 2008). Only Oak and Pinyon-Juniper showed higher recovery rates under high
fireseverity among forest types which is primarily due to rapid regeneration by resprouting in Oak
(Meng et al., 2018) and colonization by resprouting shrubs in Pinyon-Juniper (Wangler &
Minnich, 1996). The low importance of fire severity in determining post-fire vegetation growth
indicates that the variability across a single fire may be outweighed at a regional level by climate
and its proxies. It also suggests that at some sites, the impact of wildfire may be restricted to
causing tree mortality under changing climate, rather than also significantly influencing the post-
fire regeneration with its impact on seed availability (Kemp et al., 2019).
Our analysis indicated that among all the factors considered, elevation had the highest variable
importance score in predicting the LAI 10-year and 20-year post-fire. We found greater rates of
vegetation recovery in lower elevation. Less successful recovery at higher elevations is likely
associated with cooler temperatures at higher elevations for many of the forest types, and those
cool temperatures appear to still limit forest establishment and growth, even under general
warming in the region (Stevens-Rumann et al., 2018). A possible secondary reason could be soil
conditions in the mountainous terrain and slope, with a higher occurrence of steep slopes at higher
elevations than lower elevations. Slope has been shown to result in lower regeneration density
compared to shallower slopes (Lyderson & North, 2012; Kemp et al., 2016). Only Pinyon-Juniper
showed increased recovery with elevation (Fig. 5b and 6b) likely due to relief from the hot, dry
conditions at lower elevations but also possibly due to resistance to invasion that increases with
elevation in this forest type (Urza et al., 2017), suggesting that warming temperatures are having



a detrimental effect on post-fire regeneration at warmer sites, but not yet promoting post-fire
regeneration at cooler sites at all spatial scales (Harvey et al., 2016). Elevation was found to be
important in various studies of post-fire regeneration of conifer forests in the western U.S., but
with opposite directionality (Casady et al., 2010; Rother & Veblen, 2016; Vanderhoof et al., 2020).
However, Mantgem et al., (2006) reported a strongly negative correlation with seedling density of
Mixed conifer forests in the Sierra Nevada. In higher elevation forests such as Lodgepole pine,
most studies demonstrated increased recovery post-fire (e.g., Harvey et al., 2016) which contrasted
with our findings. However, modeling evidence suggests that Lodgepole pine regeneration post-
fire could experience significant declines in coming decades as a result of both increased fire
frequency (Westerling et al., 2011) and changing climatic conditions (Coops & Waring, 2011).
These findings collectively highlight that there exists a large degree of uncertainty around
individual forest type responses to post-fire climatic variability.
Our study adds to a growing body of literature emphasizing the importance of climate for post-fire
vegetation growth among different forest types (Meng et al., 2015; Buechling et al., 2016; Rother
and Veblen, 2017; Hankin et al., 2019; Vanderhoof et al., 2020). Our data suggest that high average
summer temperatures and low water availability limit the recovery of LAI 10-year and 20-year
postfire on these forest types. Drier forests such as Oak, Ponderosa pine, Douglas-fir, and Pinyon-
Juniper were strongly associated with annual precipitation and mean summer temperature, which
is consistent with Meng et al., (2015) who reported a positive relationship between five-year post-
fire NDVI values and wet season precipitation anomaly in Mixed conifers of Sierra Nevada.
Similarly, Kemp et al., (2019) found mean summer temperature to be very important indicator of
post-fire regeneration for Douglas-fir and Ponderosa pine with decreased potential for successful
regeneration under warmer summer temperatures. Our analysis also suggests that the critical



thresholds for annual precipitation and mean summer temperature are 500 mm and 15-20℃,
respectively, in these forest types. Our finding of higher sensitivity of Oak, Ponderosa pine,
Douglas-fir, and Pinyon-Juniper to annual precipitation and average summer temperature suggests
that future increases in temperature and water deficit may affect these forest types more so than
other forest types. For example, Rehfeldt et al., (2014) predicted a 50% decline in Ponderosa pine
habitat range by 2060 in response to climate change. With a trend toward warmer springs and
summers in recent decades throughout the western US (Westerling, 2006; Ghimire et al., 2012;
IPCC, 2013; Williams et al., 2021), conditions for post-fire vegetation growth and survival are
changing, as even a slight increase in water deficit on the drier sites can have adverse effects on
tree regeneration (Stevens-Rumann et al., 2018). While warming temperature has been shown to
affect the post-fire regeneration of confer forests growing at the warmer end of the species
distribution such as Douglas-fir and Ponderosa pine (Haffey et al., 2018; Kemp et al., 2019), it
could promote the rate of post-fire recovery for conifer forests growing at the colder end of the
species distribution previously limited by frozen soils, cold temperatures, and snow (Stevens-
Rumann et al., 2018; Vanderhoof et al., 2020).
Similar to LAI, our results of variable importance in random forests showed low importance of
fire severity compared to other variables in post-fire recovery of summer albedo at both time
intervals (Fig. S3). However, we noticed a difference in albedo change across fire severity classes.
For example, we found lower albedo values in low fire severity areas compared to medium and
high severity areas at both time intervals, which is associated with a greater degree of LAI recovery
in low severity areas as vegetation has lower albedo than bare areas. Moreover, lower albedo 10-
years post-fire in high severity compared to medium severity could be due to standing snags
absorbing sunlight, with it taking 5-15 years for just half of dead snags to fall (Russell et al., 2006).



We did not find significant impact of elevation on post-fire albedo change in these forest types
except for Pinyon-Juniper and Ponderosa pine, which showed decreased albedo post-fire in
response to increased LAI with elevation. As expected, climate, particularly annual precipitation,
was the major determinant of post-fire albedo change. Annual precipitation was found to be highly
associated with changes in post-fire albedo in all forest types, where increased precipitation
decreased the albedo post-fire with impact more prominent in 20-year post-fire. Annual
precipitation impacts post-fire albedo through two different mechanisms. First, increased annual
precipitation is associated with greater recovery of LAI in these forest types (Fig. 6c) where the
mid-age stands replace the initial post-fire establishments, reducing albedo (Chambers and Chapin,
2002). Second, soil moisture depends on precipitation. With greater precipitation leading to
increased soil water content, there is corresponding decrease in albedo due to darkening of soil
(Domingo et al., 2000) and an increase in leaf area within the understory during the wet season
(Thompson et al., 2004). Regarding temperature, the pattern of albedo recovery did not correspond
well with the pattern of LAI recovery at both time intervals in these forest types. Albedo is elevated
over the pre-fire condition more in the warmer part of a forest type's range even in forest types
that have a faster recovery of LAI in that warmer domain. We might expect that a higher LAI
would be associated with a lower albedo, but evidently the association is not as simple, and it
might have something to do with species composition rather than simply leaf area. Our results
point to the importance of climate patterns as a driver of post-fire summer albedo recovery through
their influence on ecological succession on the post-fire environment.
**4.4. Significance and limitations of our Analysis**
Our results should be interpreted in light of four constraints. First, the accuracy of MODIS product
algorithm is dependent on biome-specific values, which following extensive fire-caused mortality,



can introduce additional uncertainty. For instance, the use of look-up-table (LUT) for different
biomes in the MODIS fPAR/LAI algorithm can potentially lead to errors in LAI derivation in post-
fire environment if an incorrect biome classification is applied. In addition, we utilized the
recovery of MODIS LAI as an indicator of vegetation recovery. However, it is important to
acknowledge that LAI is a valuable yet imperfect indicator of vegetation change resulting from
wildfires. One significant limitation of LAI-based analysis is that it captures some of the aggregate
effects of mortality and regrowth but does not fully characterize shifted species composition and
community structure on the ground. We recognize that short-term LAI following wildfire
represents relative vegetation cover rather than a direct measure of forest regeneration. Therefore,
detailed, intensive field monitoring of vegetation structure both before and after fires can serve as
a valuable complement to LAI-based analysis (Williams et al., 2014). Additionally, incorporating
additional remote observations at the species level from the fusion of very high spatial resolution,
lidar, or hyperspectral data (Huesca et al., 2013; Polychronaki et al., 2013; Kane et al., 2014) can
further enhance the assessment. Moreover, establishing connection between field-level data and
satellite observations can enhance the interpretability of satellite observations (Hudak et al., 2007)
and offer a means to scale up ground observations to effectively characterize full landscapes.
Second, in terms of albedo, we used a 500 m MODIS albedo product which reflects a somewhat
larger area (Campagnolo et al., 2016). Each 500 m grid may in fact include a mix of burned and
unburned patches which could result in underestimation of post-fire albedo. Moreover, the
algorithm used to calculate albedo may result in an underestimation, as it might disproportionately
consider structural elements (e.g., snags and surviving trees) in the post-fire landscape. A modeling
study by Hovi et al (2019) corroborated this who reported strong link between the effective spatial
resolution of the MODIS albedo product and forest structure. Although the use of MODIS data





with its relatively low spatial resolution will miss some of the details of fine-scale spatial
variability in burn severity, land cover type and so forth (Key, 2006), MODIS data has advantages
in terms of higher temporal frequency of sampling that can be important in post-fire biophysical
dynamics (Lhermitte et al, 2010; Veraverbeke et al., 2010, 2012) and these data also have good
temporal coverage going back decades. Furthermore, higher resolution datasets on biophysical
properties are still not operationally available. Third, the quality of our results may be constrained
by the accuracy of fire severity from the MTBS product as dNBR is not a perfect metric of severity
and may struggle to capture some variations in severity (Roy et al., 2006; De Santis and Chuvieco,
2009). However, several new generation fire remote sensing products (Csiszar et al., 2014; Parks
et al., 2014; Boschetti et al., 2015) are emerging in recent years, which hold the potential for further
improvements in post-fire recovery studies. Finally, the processes driving post-fire recovery in
burned areas may vary from one location to another. The interaction among all the determinant of
post-fire forest recovery is complex and measurements of fine resolution topo-climatic variables
may not adequately explain the processers involved in forest regeneration and survival in the post-
fire environment. There are several other factors that influence post-fire regeneration that this
study did not consider but could be important like species competition (Hansen et al., 2016;
Stoddard et al., 2018), distance to seed tree (Kemp et al., 2016; Stevens-Rumann and Morgan,
2019), and other pre-fire disturbances (Buma and Wessman, 2011). The majority of the studies on
post-fire recovery presented here have attributed the slower rates of recovery to post-fire climate
conditions. To gain a comprehensive understanding of the trajectory of post-fire vegetation
recovery, future studies, in addition to topo-climatic variables, should consider physiology of
cones, seeds, and seedlings,as well as the interactions among all influencing drivers in these
settings.



Despite these limitations, by aggregating across multiple fire events in 21 different sub-ecoregions
and arraying observations along a 25-years chronosequence, our results demonstrate the spatial
and temporal variability of fire effects on post-fire environment. While forest regeneration may be
low in burned areas, it is highly variable spatially which is evident from the difference in recovery
rates between moist, cooler northern sub-ecoregions and dry, hot southern sub-ecoregions.
Understanding such variability of fire effects and vegetation in space and time is important for
comprehensive understanding of the drivers of natural regeneration and vegetation recovery in
post-fire environments (Stevens-Rumann and Morgan, 2019). Our analysis could also help
improve the modeling of post-fire recovery pathways by identifying the most important predictors
of post-fire recovery and by approximating related thresholds of response. For example, our results
suggest a full recovery of LAI in dry, low elevation forest types like Pinyon-Juniper, Ponderosa
pine, and Oak within 10 years post-fire when the annual precipitation exceeds the threshold of 500
mm and average summer temperature is ~15-20°C. A quantitative measure of primary controls is
needed if efforts to develop realistic post-fire LAI trajectories for ecohydrological modeling
studies are to be successful, as suggested by McMichael et al., (2004).
One major significance of our approach and findings is its potential to advance the land surface
models (LSMs) embedded in Earth system models (ESMs). For instance, the patterns emerged
from our data analysis could be utilized to inform model parameters that describe wildfire impacts
on biophysical properties of a landscape. A common practice in land surface modeling is to define
a set of parameter values that are relatively constant for specific biomes all over the world (for
example, Betts et al., 2007) and therefore, misses the local ecological dynamics of each biome,
weakening the model-based assessments (Myhre et al., 2005; Barnes & Roy, 2010). This holds
true in post-fire environment and is evident from this study that suggests that the parameter values





associated with biophysical, hydrological, and biogeochemical processes such as LAI and albedo
vary over space and environmental condition, even within a specific vegetation type. Therefore,
subtle changes to response functions and parameterization that govern rates of carbon, energy, and
water fluxes in relation to disturbance events can yield divergent modeled responses of ecosystems
to disturbance events. Currently, these models lack robust representations of the ecological and
biophysical consequences resulting from wildfire events (Lawrence and Chase, 2007; Williams et
al., 2009). In this research, we have quantified the post-fire changes in biophysical properties of
land surface as a function of time since fire. Modelers could use these annual values to inform the
LSMs to more accurately represent biophysical and ecological functions of severely disturbed
landscapes.
**4.5. Implications of Our Research**
There is mounting evidence of increased extreme fire incidents in the western US due to ongoing
climate change (Westerling et al., 2006; Williams et al., 2014), leading to rapid alteration and
considerable uncertainty regarding species composition (McDowell et al., 2015) and ecological
dynamics (Johnstone et al., 2016). This study provides an estimate of the effect of the post-fire
environment on vegetation and surface albedo balance of the western US. The chronosequence
data show clear patterns with time since fire for both biophysical parameters. Our results
quantitatively suggest that conifer forest ecosystems, particularly Douglas-fir and Ponderosa pine,
are more vulnerable in the drier interiors of the western US exposed to high severity fires and this
vulnerability is projected to increase in coming decades as wildfires continue to increase in severity
and size under warmer and drier climate conditions (Abatzoglou and Williams, 2016; Littell et al.,
2018). The post-fire biophysical changes documented here could be of significance for local to



regional climates, potentially eliciting feedbacks that influence regional climate change and needs
for adaptation.
**Code and Data Availability**
All of the research input data and codes supporting the results reported in this paper can be
accessed through https://doi.org/10.5281/zenodo.7927852.
**Author Contribution**
The first author conceptualized and designed the research, curated data, ran the analysis and wrote
a draft. The second author (Dr. Christopher A. Williams) provided substantial input in research
conceptualization, research framework, and polishing of the manuscript. Drs. Brendan M. Rogers,
John Rogan, and Dominik Kulakowski offered insight into the manuscript's data analysis
presentation and contributed to the draft manuscript's finalization.
**Conflict of Interest**
The authors declare that they have no known competing financial interests or personal
relationships that could have appeared to influence the work reported in this paper.



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

Limitations to recovery following wildfire in dry forests of southern Colorado and northern

New Mexico, USA, *Ecological Applications 30*, e02001, 2020.

Rodrigo, A., Retana, J., Picó, F. X.: Direct regeneration is not the only response of Mediterranean

forests to large fires, Ecology, 85, 716–729, 2004.



Knox, K. J. E., Clarke, P. J.: Fire severity, feedback effects and resilience to alternative community
states in forest assemblages, Forest Ecology and Management, 265, 47–54, 2012.
Rogers, B. M., Neilson, R. P., Drapek, R., Lenihan, J. M., Wells, J. R., Bachelet, D., Law, B. E.:
Impacts of climate change on fire regimes and carbon stocks of the U.S. Pacific Northwest,
*J. Geophys. Res. Biogeosciences 116*, 1–13, https://doi.org/10.1029/2011JG001695, 2011.
Rogers, B. M., Randerson, J. T., Bonan, G. B.: High-latitude cooling associated with landscape
changes from North American boreal forest fires, Biogeosciences, 10, 699–718,
https://doi.org/10.5194/bg-10-699-2013, 2013.
Rogers, B. M., Soja, A. J., Goulden, M. L., Randerson, J. T.: Influence of tree species on
continental differences in boreal fires and climate feedbacks, Nat. Geosci. 8, 228–234.
https://doi.org/10.1038/ngeo2352, 2015.
Rother, M. T., Veblen, T. T.: Limited conifer regeneration following wildfires in dry ponderosa
pine forests of the Colorado Front Range, Ecosphere 7, https://doi.org/10.1002/ecs2.1594,

2016.

Rother, M. T., Veblen, T. T.: Climate drives episodic conifer establishment after fire in dry
ponderosa pine forests of the Colorado Front Range, USA, Forests, 8, 1–14,
https://doi.org/10.3390/f8050159, 2017.
Roy, D.P., Boschetti, L., Trigg, S.N.: Remote sensing of fire severity: assessing the performance
of the normalized burn ratio, IEEE Geoscience and Remote Sensing Letters, 3, 112–116,

2006.

Ruefenacht, B., Finco, M., Czaplewski, R., Helmer, E., Blackard, J., Holden, G., Lister, A.,



Salajanu, D., Weyermann, D., Winterberger, K.: Conterminous US and Alaska forest type

mapping using forest inventory and analysis data, *Photogramm. Eng. Remote Sensing 74*,

1379–1388, 2008.

Russell, R. E., Saab, V. A., Dudley, J. G., Rotella, J. J.: Snag longevity in relation to wildfire and

postfire salvage logging, Forest Ecology and Management, 232, 179–187, 2006.

Salomonson, V. V., Appel, I.: Estimating fractional snow cover from MODIS using the normalized

difference   snow   index,   *Remote   Sensing   of   Environment,   89*   (3),   351–360,

https://doi.org/10.1016/j.rse.2003.10.016, 2004.

Savage, M., Brown, P. M. Feddema, J.: The role of climate in a pine forest regeneration pulse in

the southwestern United States, *Ecoscience, 3*, 310–318, 1996.

Schaaf, C. B., Gao, F., Strahler, A. H., Lucht, W., Li, X., Tsang, T., Strugnell, N. C., Zhang, X.,

Jin, Y., Muller, J., Lewis, P., Barnsley, M., Hobson, P., Disney, M., Roberts, G.,

Dunderdale, M., Doll, C., Robert, P., Hu, B., Liang, S., Privette, J. L., Roy, D.:  First

operational BRDF, albedo nadir reflectance products from MODIS, *Remote Sens. Environ.*

*83*, 135–148, 2002.

Scholze, M., Knorr, W., Arnell, N. W., Prentice, I. C.: A climate-change risk analysis for world

ecosystems, *Proceedings of the National Academy of Sciences of the United States of*

*America, 103*(35), 13116–13120, 2006.

Seastedt, T. R., Hobbs, R. J., Suding, K. N.: Management of novel ecosystems: Are novel

approaches required? *Frontiers in Ecology and the Environment, 6*, 547–553, 2008.

Shrestha, S., Williams, C.A., Rogers, B.M., Rogan, J., Kulakowski, D.: Wildfire controls on land



surface properties in mixed conifer and ponderosa pine forests of Sierra Nevada and
Klamath    mountains,    Western    US,    Agric.    For.    Meteorol.    320,    108939,
https://doi.org/10.1016/j.agrformet.2022.108939, 2022.
Shrestha, S., Williams, C.A., Rogers, B.M., Rogan, J., Kulakowski, D.: Forest Types Show
Divergent Biophysical Responses After Fire: Challenges to Ecological Modeling [Data
set], Zenodo, https://doi.org/10.5281/zenodo.7927852, 2023.
Stevens-Rumann, C. S., Kemp, K. B., Higuera, P. E., Harvey, B. J., Rother, M. T., Donato, D. C.,
Morgan, P., Veblen, T. T.: Evidence for declining forest resilience to wildfires under
climate change, Ecol. Lett. 21, 243–252, https://doi.org/10.1111/ele.12889, 2018.
Stevens-rumann, C. S., Morgan, P.: Tree regeneration following wildfires in the western US : a
review 1, 1–17, 2019.
Stoddard, M. T., Huffman, D. W., Fulé, P. Z., Crouse, J. E., Meador, A. J. S.: Forest structure and
regeneration responses 15 years after wildfire in a ponderosa pine and mixed-conifer
ecotone, Arizona, USA, Fire Ecol. 14, 1–12, https://doi.org/10.1186/s42408-018-0011-y,

2018.

Strobl, C., Boulesteix, A.-L., Zeileis, A., Hothorn, T.: Bias in random forest variable importance
measures:    Illustrations,    sources    and    a    solution,    *BMC Bioinformatics*,    *8*,    25,
https://doi.org/10.1186/1471-2105-8-25, 2007.
Thompson, C., Beringer, J., Chapin, F.S., McGuire, A.D.: Structural complexity and land-surface
energy exchange along a gradient from arctic tundra to boreal forest, J. Veg. Sci. 15, 397–
406, https://doi.org/10.1111/j.1654-1103.2004.tb02277.x, 2004.



Thompson, R. S., Shafer, S. L., Strickland, L. E., Van de Water, P. K., Anderson, K. H.: Quaternary

vegetation and climate change in the western United States: Developments, perspectives,

and prospects, *Dev. Quat. Sci*. 1, 403–426, https://doi.org/10.1016/S1571-0866(03)01018-

2, 2003.

Tsuyuzaki, S., Kushida, K., Kodama, Y.: Recovery of surface albedo and plant cover after wildfire

in a *Picea mariana* forest in interior Alaska, *Climatic Change* **93**, 517–525,

doi:10.1007/S10584-008-9505-Y, 2009.

Urza, A. K., Weisberg, P. J., Chambers, J. C., Dhaemers, J. M., Board, D.: Post-fire vegetation

response at the woodland–shrubland interface is mediated by the pre-fire community,

Ecosphere, 8, https://doi.org/10.1002/ecs2.1851, 2017.

U.S. Geological Survey.: 3D Elevation Program 30-Meter Resolution Digital Elevation Model,

2019. Assessed December 30, 2019 at https://www.usgs.gov/the-national-map-data-

delivery

Van Mantgem, P. J., Stephenson, N. L., Keeley, J. E.: Forest reproduction along a climatic gradient

in the Sierra Nevada, California, For. Ecol. Manage., 225, 391–399,

https://doi.org/10.1016/j.foreco.2006.01.015, 2006.

Vanderhoof, M. K., Hawbaker, T. J., Ku, A., Merriam, K., Berryman, E., Cattau, M.: Tracking

rates of postfire conifer regeneration vs. deciduous vegetation recovery across the western

United States, *Ecol. Appl.* 31, https://doi.org/10.1002/eap.2237, 2020.

Veraverbeke, S., Gitas, I., Katagis, T., Polychronaki, A., Somers, B., Goossens, R.: Assessing post-

fire vegetation recovery using red-near infrared vegetation indices: Accounting for

background and vegetation variability, *ISPRS J. Photogramm. Remote Sens.* 68, 28–39,



https://doi.org/10.1016/j.isprsjprs.2011.12.007, 2012, a.

Veraverbeke, S., Lhermitte, S., Verstraeten, W. W., Goossens, R.: The temporal dimension of

differenced Normalized Burn Ratio (dNBR) fire/burn severity studies: The case of the large

2007 Peloponnese wildfires in Greece, *Remote Sens. Environ.* 114, 2548–2563,

https://doi.org/10.1016/j.rse.2010.05.029, 2010.

Veraverbeke, S., Verstraeten, W. W., Lhermitte, S., Van De Kerchove, R., Goossens, R.:

Assessment of post-fire changes in land surface temperature and surface albedo, and their

relation with fire burn severity using multitemporal MODIS imagery, Int. J. Wildl. Fire 21,

243–256, https://doi.org/10.1071/WF10075, 2012, b.

Wangler, M.J., Minnich, R.A.: Fire and Succession in Pinyon-Juniper Woodlands of the San

Bernardino Mountains, California. Author(s): Michael J . Wangler and Richard A .

Minnich.    Published    by :    California    Botanical    Society    Stable    URL :

http://www.jstor.org/stable/41425166. References 43, 493–514, 1996.

Welch, K. R., Safford, H. D., Young, T. P.: Predicting conifer establishment post wildfire in mixed

conifer forests of the North American Mediterranean-climate zone, *Ecosphere, 7*,

https://doi.org/10.1002/ecs2.1609, 2016.

Westerling, A. L., Hidalgo, H. G., Cayan, D. R., Swetnam, T. W.: Warming and earlier spring

increase   Western   U.S.   forest   wildfire   activity,   *Science*,   *313*(5789),   940–943,

http://doi.org/10.1126/science.1128834, 2006.

Westerling, A. L., Turner, M. G., Smithwick, E. A. H., Romme, W. H., Ryan, M. G.: Continued

warming could transform greater Yellowstone fire regimes by mid-21st century,



*Proceedings of the National Academy of Sciences* 108, 13165–13170, https://doi.org/10.1073/pnas.1110199108, 2011.

Williams, A. P., Abatzoglou, J. T.: Recent Advances and Remaining Uncertainties in Resolving Past and Future Climate Effects on Global Fire Activity, *Current Climate Change Reports*, *2*(1), 1–14, http://doi.org/10.1007/s40641-016-0031-0, 2016.

Williams, A. P., Seager, R., Berkelhammer, M., Macalady, A. K., Crimmins, M. A., Swetnam, T. W., Trugman, A. T., Buenning, N., Hryniw, N., McDowell, N. G., Noone, D., Mora, C. I., Rahn T.: Causes and implications of extreme atmospheric moisture demand during the record-breaking 2011 wildfire season in the southwestern United States, *Journal of Applied Meteorology and Climatology 53*, 2671–2684, doi: 10.1175/JAMC-D-14-0053.1, 2014.

Williams, C. A., Collatz, G. J., Masek, J., Goward, S. N.: Carbon consequences of forest disturbance and recovery across the conterminous United States, *Global Biogeochem. Cycles, 26*(1), GB1005, doi:10.1029/2010GB003947, 2012.

Williams, C.A., Gu, H., Jiao, T.: Climate impacts of U.S. forest loss span net warming to net cooling, *Sci. Adv. 7*, 1–7, https://doi.org/10.1126/sciadv.aax8859, 2021.

Williams, C.A., Vanderhoof, M.K., Khomik, M., Ghimire, B.: Post-clearcut dynamics of carbon, water and energy exchanges in a midlatitude temperate, deciduous broadleaf forest environment, *Glob. Chang. Biol. 20*, 992–1007, https://doi.org/10.1111/gcb.12388, 2014.

Williams, M., Richardson, A.D., Reichstein, M., Stoy, P.C., Peylin, P., Verbeeck, H., Carvalhais, N., Jung, M., Hollinger, D.Y., Kattge, J., Leuning, R., Luo, Y., Tomelleri, E., Trudinger, C.M., Wang, Y. P.: Improving land surface models with FLUXNET data, *Biogeosciences 6*, 1341–1359, 2009.





Wittenberg, L., Malkinson, D., Beeri, O., Halutzy, A., Tesler, N.: Spatial and temporal patterns of

vegetation recovery following sequences of forest fires in a Mediterranean landscape, Mt.

Carmel Israel, *Catena 71*, 76–83, https://doi.org/10.1016/j.catena.2006.10.007, 2007.

Yang, J., Pan, S. Dangal, S. Zhang, B. Wang, S. Tian. H.: Continental-scale quantification of post-

fire vegetation greenness recovery in temperate and boreal North America, *Remote Sensing*

*of Environment 199*, 277–290. https://doi.org/10.1016/j.rse.2017.07.022, 2017.

Zhao, F. R., Meng, R., Huang, C., Zhao, M., Zhao, F. A., Gong, P., Yu, L., and Zhu, Z.: Long-

term post-disturbance forest recovery in the Greater Yellowstone ecosystem analyzed

using Landsat time series stack, *Remote Sensing 8*, 1–22, 2016.