# Peer review of "Divergent Biophysical Responses of Western United States Forests to Wildfire Driven by Eco-climatic Gradients"

_EGUsphere, 2023_

## Author Comment (AC1)

Summary – This manuscript tracks post-fire recovery of MODIS albedo and LAI for forested burned areas across the western U.S. The scale of the effort and its findings are important, but the Methods and Discussion both need revision, and I had concerns about how snow cover was considered and how burn severity was characterized.

**Major Comments**

Line 185 – Patterns of winter albedo and LAI will be highly dependent on the consistency of snow cover (and to a lesser degree snow depth). Grouping by snow cover conditions is briefly mentioned on line 185 but it isn't clear how or if this was controlled for or evaluated. Please provide more detail here, and control for snow cover consistency in the winter albedo results. For sites with inconsistent snow cover, producing an average D, J, F value will likely contribute quite a lot of "noise" to the findings.

Response: The reviewer is correct that snow cover can have a major effect on surface albedo, which is why albedo data products are commonly stratified into snow-covered and snow-free conditions, where a threshold of day-specific snow cover percent is adopted to sample these different conditions while ensuring sufficient data populations.  We assigned pixels to these two conditions with thresholds of less than 30% and greater than 75% snow cover on a given day and pixel for snow-free and snow-covered data populations.  This is explained in the methods at Lines 158 – 164, reading as follows.

"*We stratified the sampling of white-sky albedo by snow-free and snow-covered conditions based on the presence or absence of snow, determined at a pixel level by the MODIS daily snow cover 500 m product (MOD10A1; Salomonson and Appel, 2004). We assigned snow-free and snow-covered conditions using a threshold of less than 30% and greater than 75% snow cover. We chose these thresholds as a balance between inclusion for robust sampling and exclusion to reduce noise from pixels with an unclear mix of snow and snow-free conditions.*"

We believe that our sampling technique of averaging daily value for a pixel for all of the dates on which its snow cover was above 75% over the three month period is an adequate approach to characterize to derive the snow-covered winter albedo, establishing snow-dominated conditions while balancing the need for statistically-robust sampling. The sampling does not require a given pixel to have snow cover consistently over three months for it to contribute to the data population of snow-covered conditions.

Line 197 – as MTBS severity classes are subjectively determined per fire, they are rarely used and typically not intended to be used across multiple fires, instead authors more often use MTBS for fire perimeters, then generate dnbr and threshold burn severity classes independent from MTBS classes. The use of burn severity classes across many fires, therefore, really needs to be justified. In addition, because everything is being resampled to 500 m, why not just use MOSEV (ESSD - MOSEV: a global burn severity database from MODIS (2000–2020) (copernicus.org)) to classify burn severity?

Response: Thematic classification of burn severity in MTBS is already based on dNBR so we did not generate dNBR and burn severity classes of our own and independent of the MTBS's attribution to these classes. We agree that dNBR is not a perfect metric of severity and we have discussed this limitation in the discussion and conclusion section. We used the fire severity classes as identified by MTBS for consistency across ecoregions. We preferred MTBS data for two reasons 1) it has finer resolution and 2) it provides discrete fire severity classes. In addition, while aggregating to 500m, we selected only those 500m pixels that are at least 75% burned at high severity. This sampling technique improved our ability to sample MODIS albedo pixels that have relatively homogenous burn conditions, thus reducing the noise with an unclear mix of burned and unburned conditions and with variability in burn severity.

Discussion – the discussion is currently 15 pages long! This is way too long, especially relative to the length of the other sections, for example the Introduction is 3 pages long. Please cut this section at least in ½, I think otherwise people just aren't going to read any of it. Although the section is very nicely organized and well written, most of the paragrahs and sections could be written a lot more concisely. For example, lines 700-717 could be easily re-written to be 1-2 sentences.

Response: Thank you for pointing this out. Although we have used double line space in this manuscript, we agree with the reviewer that the discussion section is long compared to other sections. We have attempted to reduce the discussion section in the newer version of our manuscript, removing 22% of the total original word count.

**Minor Comments**

Lines 3-4 – I would disagree that efforts are lacking over larger geographic scales, please reword this sentence maybe to use the word "limited" instead. Examples to the contrary here could include:

- Yang et al. 2017 Continental-scale quantification of post-fire vegetation greenness recovery in temperate and boreal North America - ScienceDirect

- Littlefield et al. 2020 A climatic dipole drives short- and long-term patterns of postfire forest recovery in the western United States (pnas.org)

- Hislop et al. 2020 A satellite data driven approach to monitoring and reporting fire disturbance and recovery across boreal and temperate forests - ScienceDirect

Response: We changed "lacking" to "limited" and now also cite the papers that were brought to our attention by the reviewer.

Line 9 – were prescribed fires excluded from the fires? Probably more appropriate to use wildland fire as the term here.

Response: We do not include prescribed fires. Regarding the term wildland versus wildfire, we appreciate the difference and although wildland fire and wildfire are used interchangeably in

scientific literatures, we have found that wildfire is used most often, and we have followed that practice here.

Figure 1 – I would recommend changing the spruce/fir/hemlock to a different color to improve contrast with the burned area. Also consider changing wildfire to "burned area" since that is what is being mapped.

Response: Updated as recommended.

Line 125 – change "reprojected" to "resampled"

Response: Updated as recommended.

Lines 127-132 – how many fires were sampled? Total burned area by forest type? It would be helpful to include some basic numbers in a table to help clarify how much area was included in the analysis.

Response: We sampled all the wildfires that occurred between 1986 and 2017, but only those pixels that were burned with high severity in each forest type were included in post-fire biophysical dynamics analysis. We have reported total area burned by sub-ecoregions in supplementary table (Table S1)

 Line 175 – change "projection" to "resolution"

Response: Updated as recommended.

Line 197 – In line 109 the authors specify that they only used high severity MTBS? How was 3 classes of burn severity present then? How were mixed severity classes attributed during the resampling process? Please clarify in the text.

Response: We used only high fire severity to quantify the biophysical dynamics in post-fire environment. However, in a separate analysis of attribution of recovery, we used MTBS fire severity classes (low, medium, and high) to examine how these classes determine the recovery of LAI and albedo 10 and 20 years after fire. For this purpose, we did not use our own classification, rather we used fire severity classes strictly defined by MTBS.

Line 213 – Specify what parameter values were selected

Response: The following line was updated in the revised manuscript:

"*We created four RF models with 500 binary decision trees for each forest type (one for each time horizons for both LAI and albedo)*"

Figure 2- Is this supposed to be "Coastal sage"? Also, where sub-ecoregions are used in multiple panels, please try to keep the color the same for that sub-ecoregion.

Response:  The reviewer is correct. We have changed it to "Coastal sage".

We believe that the reader might find it difficult to read the figures using the same color for a sub-ecoregion (n=21) in all forest types. Please see the figure below for your reference. The purpose of using distinct colors for sub-ecoregions (separate legend) in each forest type in our manuscript is to increase the readability of the graph.

[Figure]

Figure 4 – Some of the lines don't seem to extent through -3, what is going on there? Also some of the forest types, like panel b and e, show dramatic variability in winter albedo pre-fire, any idea what is going on there?

Response: We are lacking snow-covered albedo samples during pre-fire years in some of these forest types causing it not to extent through -3. The dramatic variability is likely due to noise associated with smaller sample size and variability of snow cover in some extent which is stated in the discussion section.

Lines 143-144 – I wonder if this is because of using MTBS as well as resampling burn severity from mixed MTBS severity classes.

Response: We are not sure how this comment is related to the statement in lines 143 – 144 that describes the algorithm used in MODIS LAI product.

---

## Author Comment (AC2)

Major comments

Overall, this is a good and important analysis, and a well written paper. However, the discussion is too long. This could mostly be reduced by reducing speculation over variables that were not measured by this remote sensing study, projections about climate change and climate resilience, and discussion of modeling.

Response: We appreciate the comment from the reviewer. We have reduced the discussion by 22% as suggested by the reviewer.

Title: I think the context for modeling is covered well in the discussion but not important enough to elevate to the title. However, I do think "western US forests" is central and important enough to elevate to the title as these results may be specific to this geography. Suggest something like: "Divergent biophysical responses of Western U.S. forests to wildfire driven by ecoclimatic gradients"

Response: We thank the reviewer for suggesting this alternative title, which we like and have adopted in the revised version.

Under hypothesis 3, you addressed potential issues with collinearity of covariates but what about response variables LAI and albedo? Are they correlated? Would this matter?

Response: We believe that LAI and albedo are correlated as LAI is one of the variables in albedo algorithm. However, this study demonstrated that the recovery of albedo is faster than that of LAI suggesting that the post-fire recovery of albedo is driven by multiple factors in addition to early regeneration of vegetation. Moreover, our past study (Shrestha et al., 2022) of postfire biophysical dynamics in mixed conifer and ponderosa pine in Sierra Nevada and Klamath mountains showed unique relationships (different slopes and intercepts) even within same forest type which indicates LAI is not the only variable affecting albedo. Understanding the drivers of variability in response in postfire environment is one of the main points of this manuscript.

Minor comments

89 – 'failed' implies this was their goal, which it may not have been

Response: Good point, we edited this to state that the studies did not examine this feature, now reading as, "*Moreover, such studies did not examine how their results scale up to multiple fire events across broad regions*".

179 – 'yearly' to summer averages (this is confusing)
Response: We replaced "yearly" with "summer-season".

245 – edit to 'high elevation, wet'

Response: Updated as recommended.

271-272 – couple of typos.  Suggest 'Similarly, there were significant regional differences in the timing and magnitude of peak albedo(?) for a given forest type group.'
Response: Updated as recommended.

276 – new paragraph for winter
Response: Updated as recommended.

279 – I don't think snow variability is 'noise' in albedo per se.  it is snow signal.
Response: This is a good catch.  We edited the statement to read "*We observed greater inter-annual variability in the timeseries of post-fire winter albedo likely related to variability in snow cover and also a smaller signal-to-noise ratio associated with smaller sample sizes.*"

322 – 'elevation' to 'enhancement' as is confusing in geographical context.  Also 'larger' to 'greater' in that sentence.
Response: We appreciate the close, careful read and the constructive suggestions.  We changed "elevation" to "increase".

326 – are these anomalies evidence of state change? See again in Discussion at 408 to 409. Perhaps these are different species?  Early successional.  OK you get to this in 442-451
Response: Indeed, this is discussed in discussion section.

338 – 'see' to 'experience' or 'exhibit' and 'elevation' to enhancement again (and anywhere else)
Response: Again, we appreciate the helpful suggestions.  Updated as recommended.

353 – 'time events' to 'time horizons'?  again in 362 and elsewhere.
Response: Updated as recommended.

413 – soils weren't included in the analysis directly.  Indirectly?
Response: The reviewer is correct.  Our study did not include soil assessment.  We had noticed that several other studies report that soil properties influence the trajectory of post-fire vegetation growth and speculated that this could have a role. Nonetheless, we have deleted the statement from the revised manuscript.

420 – 'the decline'?
Response: Updated as recommended.

454 – moisture to moister or wetter
Response: Updated as recommended.

481 – add 'presumably' since you didn't make ground measurements
Response: Updated as recommended.

501-502, again, 'presumably'
Response: Updated as recommended.

523 – the 2014 ref. covers snags, not the 2012.
Response: Thank you for pointing this out. Updated in the revised manuscript.

543-546 – aren't you contradicting yourself?  You say fire severity is not important 'by contrast'
but then explain how it describes important differences.
Response: Our random forest results showed lowest importance of fire severity in explaining the
post-fire recovery of LAI and albedo relative to other variables. Despite its low importance, we
found differences in recovery rate across fire severity classes. Here, low importance does not
necessarily mean no predictive power.  We added a clarifying clause to the second sentence as:
"*In contrast, our analysis suggested fire severity was of relatively low importance relative to
other variables considered (Fig. S2). Despite being of lesser importance, we found that higher
rates of post-fire recovery were associated with low severity fire and lowest recovery rates were
associated with high fire severity.*"

627 – regarding the Domingo reference and this idea: the lowering of albedo with soil moisture
has been shown in drylands, I am skeptical it applies in forests, not sure this reference applies.
Response: We appreciate the reviewer's point.  This effect might apply more in open forest, low
canopy cover conditions where background soils receive direct insolation. It is generally
accepted that surface albedo decreases with increasing soil moisture and simple relationship
based on this knowledge have been mostly used in climate models. We have replaced the
reference with *Montes-Helu et al., 2009*. This research was carried out in the Ponderosa pine
forest of northern Arizona.  It now reads: "*With greater precipitation leading to increased soil
water content, we could expect a corresponding decrease in albedo due to darkening of soil
particularly in open canopy conditions where the soil received direct radiation (Montes-Helu et
al., 2009). Furthermore, an increase in leaf area within the understory during the wet season
could have a similar effect, as reported in Thompson et al. (Thompson et al., 2004).*"

679 – what about scorching the seed bank in the soil?
Response: It certainly affects post-fire vegetation recovery and we have added this to a list of
several other factors that this study did not cover. To name a few, we included distance to seed
tree, species competition, pre-fire disturbances, and physiology of cones, seeds, and seedlings.

727 – here are earlier, can you briefly clarify how you are defining vulnerability in this case?

Response: We had used the term "vulnerability" to connote slower, more limited recovery of a species post-fire.  Now we revised this to be more specific as, "*Our results show that conifer forest ecosystems, particularly Douglas-fir and Ponderosa pine, are slower to recover post-fire, which may indicate they face greater risks from the projected increase in fire severity and frequency as forecasted for drier interiors of the western US (Abatzoglou and Williams, 2016; Littell et al., 2018).*".